# Antibodies to Commonly Circulating Viral Pathogens Modulate Serological Response to Severe Acute Respiratory Syndrome Coronavirus 2 Infection

**Protim Sarker** [1,*], **Evana Akhtar** [1], **Sharmin Akter** [1], **Sultana Rajia** [1], **Rakib Ullah Kuddusi** [1], **Razu Ahmed** [1], **Md. Jakarea** [1], **Mohammad Zahirul Islam** [2], **Dewan Md Emdadul Hoque** [3], **Shehlina Ahmed** [4] **and Rubhana Raqib** [1]

1   Infectious Diseases Division, International Centre for Diarrhoeal Disease Research (icddr, b), Dhaka 1212, Bangladesh
2   Embassy of Sweden in Dhaka, Dhaka 1212, Bangladesh
3   United Nations Population Fund (UNFPA) Bangladesh, Dhaka 1207, Bangladesh
4   Foreign Commonwealth & Development Office (Bangladesh), Dhaka 1212, Bangladesh
*   Correspondence: protim@icddrb.org

**Abstract:** The purpose of this study was to determine the seropositivity of circulating viral pathogens and their association with severe acute respiratory syndrome coronavirus 2 (SARS-CoV-2) seropositivity. In a cross-sectional design, inhabitants (aged 10–60 years) of the slum and surrounding non-slum areas of Dhaka and Chattogram Metropolitan cities in Bangladesh were enrolled from October 2020 to February 2021. Antibodies to SARS-CoV-2, influenza B, parainfluenza, respiratory syncytial virus (RSV), human coronavirus HKU1 (HCoV-HKU1), dengue and chikungunya viruses were determined in plasma. The association of SARS-CoV-2 seropositivity with seropositivity to other viruses was assessed using the multi-variate logistic regression model. Seroprevalence of SARS-CoV-2, influenza B, RSV, dengue, chikungunya, HCoV-HKU1 and the parainfluenza virus were 68.3%, 98%, 50.0%, 16.5%, 15.5%, 3.36% and 0.0%, respectively. Individuals seropositive for RSV had lower odds (OR = 0.60; 95% CI= 0.49, 0.73) of SARS-CoV-2 seropositivity compared to RSV-seronegative individuals. Conversely, higher odds of SARS-CoV-2 seropositivity were observed in participants seropositive for dengue (OR= 1.73; 95% CI = 1.14, 2.66, only in slum) or chikungunya (OR = 1.48; 95% CI = 1.11, 1.95) compared to their seronegative counterparts. The study findings indicated that exposure to vector-borne virus dengue or chikungunya enhance, while antibodies to respiratory virus RSV decrease, the serological response to SARS-CoV-2.

**Keywords:** COVID-19; chikungunya; dengue; respiratory syncytial virus; seroprevalence

## 1. Introduction

Interaction between co-circulating respiratory viruses manipulates the epidemiological pattern of respiratory infections. Respiratory viruses such as influenza virus A and B, parainfluenza virus, respiratory syncytial virus (RSV), rhinovirus, adenovirus and human coronavirus HKU1 circulate commonly in the environment [1,2], and share symptoms with COVID-19, e.g., cough, headache, sore throat, fatigue, fever, muscle or body aches. Respiratory viruses may co-infect COVID-19 patients [3–10], and exacerbate the disease [11,12]. On the other hand, in patients with respiratory symptoms or influenza-like illness (ILI) in Northern Italy, detection of SARS-CoV-2 was increased with the decrease in the rates of other respiratory viruses [13,14].

There is a frightening upsurge of mosquito-borne viral infections including dengue and chikungunya, especially in tropical/subtropical urban areas [15,16], which have overlapping symptoms such as a fever, headache, rash, and muscle or body aches with SARS-CoV-2 infection. Since the first outbreak in 2000, there were several dengue outbreaks of different

frequencies and magnitudes every year in Bangladesh, with the largest one occurring in 2019. This incident peaked from July to October and caused >100,000 infections and 179 deaths in the country, especially in metropolitan cities [17–20]. Climate changes (e.g., increased rainfall, humidity, and temperature), along with rapid and unplanned urbanization facilitating mosquito breeding and viral transmission along with suboptimal vector-control programs, have contributed to recent surges of dengue cases as well as the first outbreak of chikungunya in 2017 in Bangladesh [21,22]. SARS-CoV-2 infection has been postulated to have both positive and inverse associations with dengue infection [23,24]; however, no association with chikungunya has been found [24].

We have recently shown that SARS-CoV-2 seropositivity was associated with limited years of formal education, lower income, female sex, overweight, diabetes and heart disease, while an inverse association was observed with the regular wearing of masks, washing hands and previous vaccination with BCG [25]. Acquired immunity to commonly circulating viruses may also affect the immune response to SARS-CoV-2 infection, and consequently influence the diagnostic approaches and vaccine-induced immunity [23,26].

In the present study, we aimed to evaluate whether antibodies to influenza virus B, parainfluenza virus, RSV, HCoV-HKU1, dengue virus and chikungunya virus have any association with SARS-CoV-2 seropositivity in the urban slum and surrounding non-slum areas of Bangladesh. Serological response to SARS-CoV-2 was found to be positively associated with antibodies to vector-borne virus dengue or chikungunya, and inversely associated with antibodies to respiratory virus RSV.

## 2. Materials and Methods

### 2.1. Study Design, Setting, Selection of Households and Participants

The present study is part of a community-based cross-sectional serosurvey of COVID-19, where around 3200 participants were enrolled from the large slums and adjacent non-slum areas of Dhaka and Chattogram Metropolitan cities in Bangladesh during a period which ran from October 2020 to February 2021 [25]. An Urban Health Demographic Surveillance System (UHDSS) of icddr,b monitors a dynamic cohort of a population of over 125,000 in selected urban slum areas of Dhaka to collect longitudinal health and demographic data. Furthermore, it monitors primary health care services provided by NGOs and local bodies. Korail and Mirpur slums were selected from the Dhaka North City Corporation, and Dholpur slum from the Dhaka South City Corporation. The Shaheed Lane and Akbar Shah Kata Pahar slums were selected from the Chattogram City Corporation based on the slum census report of 2014.

In the Dhaka slums, clusters (consisting of five to 30 households) were selected randomly using the UHDSS sampling frame. For the Chattogram slums, area maps were prepared by physical verification, where areas five times larger than sample households in each slum were mapped, and areas were subsequently divided into clusters. Households were selected randomly from each cluster of Chattogram slums. The number of households selected per slum was proportional to the population size of the slum. Mapping of the non-slum areas adjacent to each slum in Dhaka and Chattogram was performed by physical verification, as was done for the Chattogram slum areas. The non-slum area was selected to include middle-class apartments having one or no security guard in the building. All members of the selected households meeting the inclusion criteria were invited to participate. Study inclusion criteria included participants of either sex, aged 10–17 years (adolescents) or 18–60 years (adults), willing to provide informed consent and donate blood.

In the present study, the seroprevalence of commonly circulating viruses, including influenza virus B, parainfluenza virus, RSV, HCoV-HKU1, dengue virus and chikungunya virus was determined in a subsample ($n$ = 2012 in overall population, $n$ = 1099 in slum population, $n$ = 913 in non-slum population).

## 2.2. Collection of Data

A structured questionnaire-based survey was carried out to collect socio-demographic data (e.g., age, sex, education, occupation, monthly family income) from the selected households in the study areas. Data were collected from the head of the household or an adult member. Information on morbidity, whether current or in the past 6 months and related to COVID-like symptoms, preventive behavior (e.g., mask wearing, hand washing) practiced and chronic health conditions (e.g., diabetes, hypertension, stroke, heart disease, kidney disease, cancer) were collected from all enrolled individuals. COVID-19 vaccination in Bangladesh started after the study period, and thus information on vaccination status was not a part of this questionnaire. After completion of the interview, anthropometric data were collected: height was measured twice using a free-standing portable stadiometer (seca 217, Humburg, Germany), weight was recorded using a digital weighing scale (Camry-EB9063, HongKong, China), and body mass index (BMI) was calculated.

## 2.3. Collection, Processing and Storage of Specimen

Venous blood was collected from each participant and transferred into a trace element-free sodium heparin tube (Greiner Bio-One GmbH, Kremsmünster, Austria) and a K2-EDTA spray-coated hematology tube (Becton Dickinson, East Rutherford, NJ, USA, 07417-1880). Blood tubes were transported to the icddr,b Laboratory within three hours of collection by maintaining the cold chain. Plasma was separated from heparinized blood by centrifugation at $2000 \times g$ for 10 min, and stored at $-80\ ^{\circ}\text{C}$ in aliquots until analysis.

## 2.4. Assessment of SARS-CoV-2 Specific Antibodies

Antibodies (both IgG and IgM) to THE nucleocapsid (N) antigen of SARS-CoV-2 were determined on automated immunoassay analyzer, Cobas e601 (Roche Diagnostics GmbH, Mannheim, Germany) based on electrochemiluminescence immunoassay (ECLIA) using Elecsys Anti SARS-CoV 2 immunoassay kit (Roche Diagnostics). Based on the antibody cut-off index (COI), the serological response to SARS-CoV-2 is categorized as reactive (COI $\geq$ 1.0, seropositive) and non-reactive (COI < 1.0, seronegative).

## 2.5. Assessment of Antibodies to Other Viral Pathogens

IgG antibodies to RSV, HCoV-HKU1, influenza B virus, parainfluenza virus, dengue virus and chikungunya virus were determined in plasma by utilizing commercial ELISA kits (MyBioSource, Inc., San Diego, CA, USA). Positive, negative and in some cases equivocal results were interpreted by following the manufacturer's instructions. For RSV, HCoV-HKU1 and parainfluenza virus, the cut-off value was calculated by averaging the negative control value and adding 0.15; a value above the cut-off is recorded as positive, and below the cut-off as negative. For influenza B virus, a range of $\pm 20\%$ around the optical density (OD) value of the cutoff standard was interpreted as equivocal; an OD value above the range was defined as positive, and an OD value below the range was viewed as negative. For dengue and chikungunya virus-specific antibodies, the standard units for result interpretation were calculated by dividing the absorbance of specimen by that of calibrator multiplied by 10. A standard unit <9 was interpreted as negative, 9–11 interpreted as borderline positive or equivocal and >11 as positive.

## 2.6. Complete Blood Count (CBC)

CBC was performed from blood collected in a hematology tube using an automated hematology analyzer (XN-1000, Sysmex Corporation, Kobe, Japan).

## 2.7. Data Analysis Plan

The demographic data are expressed as mean $\pm$ standard deviation (for age only), and number with percentage. An independent sample *t* test or Pearson's Chi-square test was used to analyze the difference in demographic variables between slum and non-slum inhabitants. The prevalence of seropositivity was expressed as a percentage with 95% confi-

dence intervals (CIs). A design-adjusted Pearson's Chi-square test was used to determine the difference in seroprevalence of the virus panel (SARS-CoV-2, RSV, dengue, chikungunya, HCoV-HKU1) between slum and non-slum, and adult and adolescent categories. A weighted analysis was performed using the number of clusters from the selected areas, as well as selected participants from the clusters, to better reflect the current seroprevalence of SARS-CoV-2 among RSV, dengue- or chikungunya-seropositive and seronegative participants. The multi-variate logistic regression model was used to know the effect of seroprevalence of each virus (exposure variable) on SARS-CoV-2 seroprevalence (outcome variable). Sociodemographic factors, chronic diseases and the preventive behavior practiced were fitted in the model to evaluate their effect on the estimate of the model. Based on this evaluation, sex (categorical), age (continuous), years of education (categorical), occupation (categorical), family income (categorical), and BMI (categorical) were kept in the model as covariates, regardless of their *p* value. Multivariate regression analysis using generalized linear model was performed to estimate the mean difference in each CBC parameter (blood cell count) between seropositive and seronegative groups of viruses. The model was adjusted by age, sex, years of education, occupation, family income and BMI. All the *p* values <0.05 were considered as significant. All the analyses were performed using Stata 15 (StataCorp, LP, College Station, TX, USA) and SPSS 22 (IBM, Armonk, NY, USA).

## 3. Results

### 3.1. Demography

The demographic features of the selected population (*n* = 2012) in slums and non-slums are given in Table 1. The average age of the selected population was 31 years (range 10–57 years). The overall proportions of male and female were 42.2% and 57.8%, respectively. The proportion of people with no or only primary education is significantly higher in slums, and the number of people with 6–15 years of education is higher in non-slum area. Self-employment was significantly higher in slums, and proportion of homemakers was higher in the non-slum households. The lower income population is predominantly found in slums, and the higher income population resides in the non-slum area. The percentage of underweight and overweight population was 18.6% and 39.8%, respectively, with significantly higher proportion of underweight population being found in slums and overweight population in non-slum areas.

**Table 1.** Demographic characteristic of the study participants.

| Variables | | Overall (*n* = 2012) | Slum (*n* = 1099) | Non-Slum (*n* = 913) | *p* Value |
|---|---|---|---|---|---|
| Age, years | | 31.48 ± 16.23 | 29.65 ± 15.58 | 33.67 ± 16.72 | <0.001 |
| Sex | Male | 850 (42.2%) | 487 (44.3%) | 363 (39.8%) | 0.040 |
| | Female | 1162 (57.8%) | 612 (55.7%) | 550 (60.2%) | |
| Education | No education | 378 (18.8%) | 336 (30.6%) | 42 (4.60%) | <0.001 |
| | 1–5 years | 564 (28.0%) | 445 (40.5%) | 119 (13.0%) | <0.001 |
| | 6–10 years | 665 (33.1%) | 281 (25.6%) | 384 (42.1%) | <0.001 |
| | 11–15 years | 405 (20.1%) | 37 (3.37%) | 368 (40.3%) | <0.001 |
| Occupation | Service | 318 (15.8%) | 189 (17.2%) | 129 (14.1%) | 0.342 |
| | Self-employed | 199 (9.89%) | 168 (15.3%) | 31 (3.40%) | 0.001 |
| | Business | 163 (8.10%) | 73 (6.64%) | 90 (9.86%) | 0.214 |
| | Housewife | 545 (27.1%) | 233 (21.2%) | 312 (34.2%) | 0.002 |
| | Unemployed | 236 (11.7%) | 157 (14.3%) | 79 (8.65%) | 0.110 |
| | Student | 551 (27.4%) | 279 (25.4%) | 272 (29.8%) | 0.865 |
| Monthly income, taka | <20,000 | 360 (17.9%) | 340 (30.9%) | 20 (2.19%) | <0.001 |
| | 20,000–40,000 | 798 (39.7%) | 624 (56.8%) | 174 (19.1%) | <0.001 |
| | 40,000–70,000 | 528 (26.2%) | 135 (12.3%) | 393 (43.0%) | <0.001 |
| | >70,000 | 326 (16.2%) | 0 | 326 (35.7%) | <0.001 |

**Table 1.** *Cont.*

| Variables | | Overall (*n* = 2012) | Slum (*n* = 1099) | Non-Slum (*n* = 913) | *p* Value |
|---|---|---|---|---|---|
| BMI | Underweight | 374 (18.6%) | 281 (25.6%) | 93 (10.2%) | <0.001 |
| | Normal | 837 (38.8%) | 483 (44.0%) | 354 (38.8%) | 0.120 |
| | Overweight | 801 (39.8%) | 335 (30.5%) | 466 (51.0%) | <0.001 |
| COVID-19 like symptoms | | 810 (40.3%) | 392 (35.7%) | 418 (45.8%) | <0.001 |

BMI: body mass index. Data presented as mean ± SD or number (percent). Independent sample *t* test or Pearson's Chi-square test was used to estimate the *p* value.

### 3.2. Seroprevalence of SARS-CoV-2 and Other Circulating Viruses

The overall seroprevalence of SARS-CoV-2 in the selected population from October 2020 to February 2021 was 68.3%, with a significantly higher prevalence being found in the slum (73.4%) compared to the non-slum population (62.2%) (Table 2). About 50% of the population was positive for RSV; the positivity was significantly higher in slums than the non-slum neighborhoods (60.2% vs. 37.6%; *p* < 0.001). Overall seropositivity of dengue was 16.5% and that of chikungunya was 15.5%, and both were equally distributed in slum and non-slum areas (Table 2). Seropositivity of SARS-CoV-2, dengue and chikungunya were significantly higher in adult than adolescent participants, whereas RSV seroprevalence was similar in both age groups (Table 2). Only 3.4% of people were seropositive for HCoV-HKU1 (Table 2). About 98% of the participants were seropositive for influenza B (total), while all participants were seronegative for the parainfluenza virus. Therefore, seropositivity of HCoV-HKU1, influenza B virus and parainfluenza virus were not considered for association analysis with SARS-CoV-2 seroprevalence.

**Table 2.** Seroprevalence of SARS-CoV-2, RSV, HCoV-HKU1, dengue and chikungunya viruses among the study participants.

| | Overall (*n* = 2012) | Slum vs. Non-Slum | | | Adult vs. Adolescent | | |
|---|---|---|---|---|---|---|---|
| | | Slum (*n* = 1099) | Non-Slum (*n* = 913) | *p* Value | Adult (*n* = 1515) | Adolescent (*n* = 497) | *p* Value |
| SARS-CoV-2 | 1375 (68.3%) | 807 (73.4%) | 568 (62.2%) | <0.001 | 1064 (70.2%) | 311 (62.6%) | 0.001 |
| RSV | 1005 (50.0%) | 662 (60.2%) | 343 (37.6%) | <0.001 | 750 (49.5%) | 255 (51.3%) | 0.485 |
| HCoV-HKU1 | 67 (3.36%) | 42 (4.0%) | 25 (2.68%) | 0.116 | 44 (3.12%) | 23 (3.92%) | 0.009 |
| Dengue | 332 (16.5%) | 171 (15.6%) | 161 (17.6%) | 0.212 | 183 (18.7%) | 49 (9.86%) | <0.001 |
| Chikungunya | 312 (15.5%) | 169 (15.6%) | 143 (15.4%) | 0.927 | 260 (17.1%) | 52 (10.6%) | 0.007 |

SARS-CoV-2: severe acute respiratory syndrome coronavirus 2; RSV: respiratory syncytial virus; HCoV-HKU1: human coronavirus HKU1. Data presented as number (percent). Pearson's Chi-square test was used to estimate the *p* value.

### 3.3. Association between Seropositivity of SARS-CoV-2 and RSV/Dengue/Chikungunya

RSV-seropositive individuals had lower seroprevalence of SARS-CoV-2 compared to RSV-seronegative individuals in the overall population. This was the case in both slum and non-slum populations, as well as in both adult and adolescent groups. (Table 3). Again, the odds of participants being seropositive for SARS-CoV-2 among the RSV-seropositive participants was significantly lower compared to the RSV-negative participants (Figure 1). In contrast, among dengue-seropositive slum participants, seroprevalence of SARS-CoV-2 was much higher than among dengue-seronegative participants (80.1% vs. 67.2%); no difference was observed in the non-slum population (Table 3). The odds of SARS-CoV-2 seropositivity in the slum area was 1.73 times higher in dengue-seropositive participants than -seronegative participants (Figure 1). When stratified by age, no association was observed between dengue and SARS-CoV-2 seropositivity. Chikungunya-seropositive individuals had higher prevalence (Table 3) and 1.48 times higher odds of SARS-CoV-2 seropositivity compared to chikungunya-seronegative individuals in the overall population (Figure 1). The odds ratio of SARS-CoV-2 seropositivity was higher in chikungunya-seropositive than seronegative adults; no difference was observed among adolescents. Chronic diseases (diabetes, stroke and hypertension, only in adult participants) and behavioral practice (hand washing and mask wearing) had no effect on these associations, as these factors did not influence the estimates of the regression model.

**Table 3.** Weighted seroprevalence of SARS-CoV-2 among RSV, dengue- or chikungunya-seropositive or -seronegative participants.

| | | Overall (*n* = 2012) | Slum (*n* = 1099) | Non-Slum (*n* = 913) | Adult (*n* = 1515) | Adolescent (*n* = 497) |
|---|---|---|---|---|---|---|
| RSV | Seropositive | 61.3 (57.9, 64.6) | 65.4 (61.2, 69.4) | 53.9 (48.0, 59.7) | 64.2 (60.3, 67.9) | 53.6 (46.9, 60.1) |
| | Seronegative | 72.4 (68.5, 75.9) | 76.9 (71.7, 81.4) | 69.6 (64.2, 74.5) | 74.0 (69.7, 78.0) | 67.2 (58.9, 74.6) |
| Dengue | Seropositive | 73.2 (67.1, 78.5) | 80.1 (72.8, 85.7) | 66.6 (56.7, 75.1) | 74.2 (67.7, 79.9) | 67.9 (51.7, 80.6) |
| | Seronegative | 65.1 (62.2, 67.8) | 67.2 (63.6, 70.7) | 62.7 (58.2, 66.9) | 67.6 (64.3, 70.7) | 58.6 (53.0, 63.9) |
| Chikungunya | Seropositive | 75.3 (68.8, 80.8) | 79.4 (71.7, 85.4) | 71.7 (61.3, 80.2) | 76.4 (69.3, 82.2) | 68.9 (51.9, 82.0) |
| | Seronegative | 67.1 (64.3, 69.7) | 70.9 (67.4, 74.1) | 62.3 (57.8, 66.6) | 69.0 (65.9, 72.0) | 61.9 (56.4, 67.2) |

SARS-CoV-2: severe acute respiratory syndrome coronavirus 2; RSV: respiratory syncytial virus. Data presented as percentage prevalence (95% CI).

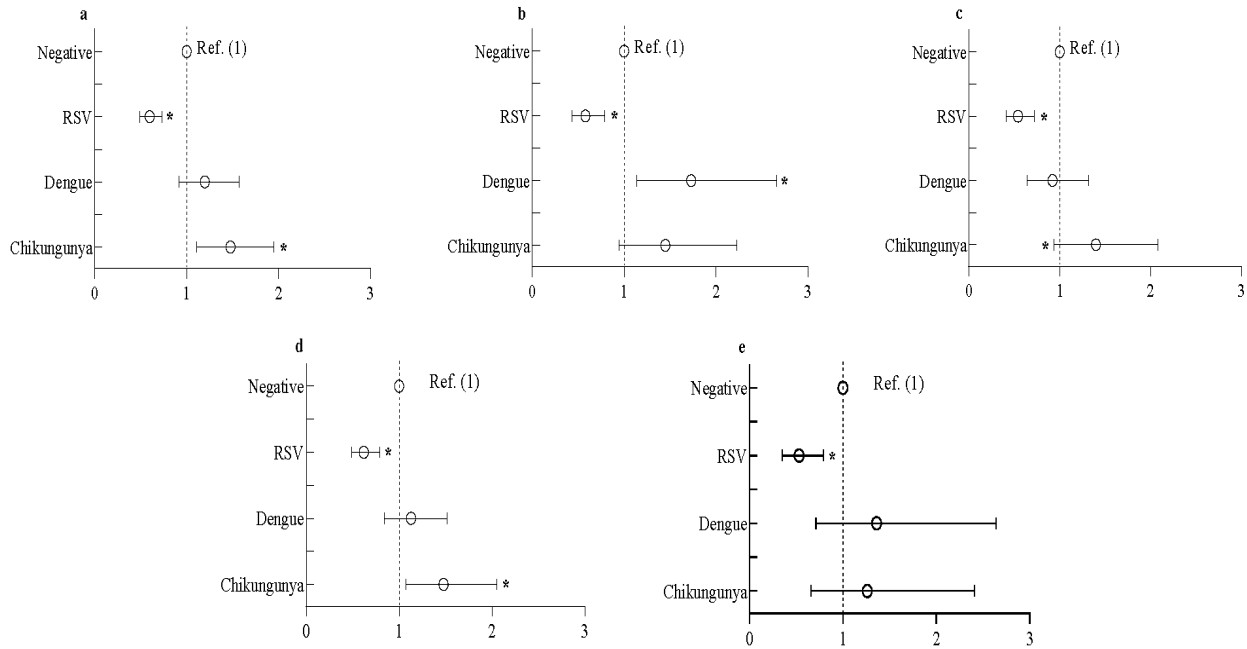

**Figure 1.** Odds of seropositivity to SARS-CoV-2 among RSV-, dengue- or chikungunya-seropositive participants. Data represents (**a**) overall study population, (**b**) urban slum population, (**c**) neighboring non-slum population, (**d**) adult population and (**e**) adolescent population. Data presented as odds ratio with 95% CI. multivariate logistic regression was used to estimate the odds ratio. The regression model was adjusted by sex (categorical), age (categorical), years of education (categorical), occupation (categorical), family income (categorical), and BMI (categorical). SARS-CoV-2: severe acute respiratory syndrome coronavirus 2; RSV: respiratory syncytial virus; BMI: body mass index. * denotes significant association.

### 3.4. Blood Cell Counts in Seropositive vs. Seronegative Groups of SARS-CoV-2/RSV/Dengue/Chikungunya

CBC analysis revealed that monocyte, eosinophil and basophil counts were significantly higher, while the platelet count was lower in the SARS-CoV-2-seropositive group compared to the seronegative group (Table 4). Chikungunya-seropositive participants also showed a significantly lower platelet count, but higher neutrophil and monocyte counts, than seronegative participants. The number of monocyte and eosinophil in RSV-seropositive cases and eosinophil in dengue-seropositive cases were significantly lower compared to their seronegative counterparts. However, no difference in platelet numbers between seropositive and seronegative groups were observed for RSV and dengue (Table 4).

**Table 4.** Mean difference in blood cell counts between seropositive and seronegative groups for different viruses.

| Blood Cells | SARS-CoV-2 | | RSV | | Dengue | | Chikungunya | |
|---|---|---|---|---|---|---|---|---|
| | β (95% CI) | *p* Value | β (95% CI) | *p* Value | β (95% CI) | *p* Value | β (95% CI) | *p* Value |
| Neutrophils | 0.06 (−0.80, 0.92) | 0.892 | −0.10 (−0.91, 0.71) | 0.802 | −0.31 (−1.38, 0.76) | 0.570 | 1.24 (0.15, 2.34) | 0.026 |
| Lymphocytes | 0.02 (−0.74, 0.79) | 0.955 | −0.45 (−1.15, 0.25) | 0.206 | 0.54 (−0.41, 1.49) | 0.265 | −0.29 (−1.26, 0.69) | 0.564 |
| Monocytes | 0.43 (0.30, 0.55) | <0.001 | −1.23 (−1.33, −1.12) | <0.001 | −0.06 (−0.22, 0.10) | 0.486 | 0.27 (0.12, 0.43) | 0.001 |
| Eosinophils | 0.70 (0.30, 1.09) | <0.001 | −1.56 (−1.93, −1.19) | <0.001 | −0.59 (−1.09, −0.09) | 0.020 | −0.08 (−0.57, 0.41) | 0.751 |
| Basophils | 0.02 (0.002, 0.03) | 0.024 | 0.01 (−0.001, 0.03) | 0.060 | 0.001 (−0.02, 0.02) | 0.891 | 0.01 (−0.01, 0.02) | 0.290 |
| Platelet | −18.6 (−27.7, −9.49) | <0.001 | 5.42 (−3.22, 14.1) | 0.219 | 9.68 (−1.73, 21.1) | 0.096 | −41.7 (−53.1, −30.2) | <0.001 |

SARS-CoV-2: severe acute respiratory syndrome coronavirus 2; RSV: respiratory syncytial virus. Multivariate regression analysis was performed using a generalized linear model to estimate the mean difference. The model was adjusted by age, sex, years of education, occupation, family income and BMI.

## 4. Discussion

The present study demonstrates that SARS-CoV-2 was highly seroprevalent (68.3%) in the urban population in Bangladesh between October 2020 and February 2021. About 50% of the study population was seropositive for RSV, and SARS-CoV-2 seropositivity was lower in RSV-seropositive patients compared to seronegative individuals. On the other hand, SARS-CoV-2 seropositivity were higher in dengue- and chikungunya-seropositive than -seronegative participants.

RSV seropositivity in the present study was observed in about 50% of the study population. Infection with RSV occurs frequently in all age groups [27–29], although the clinical burden is much higher in infants and young children who experience recurrent hospitalization [30–32]. Studies aiming to determine the transmission of RSV within families demonstrated that all age groups are infected with RSV at considerable frequency. Indeed, in majority of the cases, RSV was introduced into the family by older siblings or adult members [33–36]. In line with these findings, we have shown that 49.5% adults were seropositive for RSV, and the rate was similar to that of adolescents (51.3%). Using mathematical simulations, it was shown earlier that common cold infections are less frequent during flu seasons due to transient immune-mediated interference [37]. In the present study, we have demonstrated that individuals seropositive for RSV, irrespective of being adult or adolescent and living in slum or non-slum households, are less likely to develop antibodies to SARS-CoV-2. However, in a cross-sectional design, seroprevalence of common viral pathogens gives no information about when exactly people were exposed to different viral pathogens and generated antibodies against them, and thus it is difficult to speculate that antibodies to RSV conferred protection against SARS-CoV-2. Besides, Loos et al. recently reported that immune responses to influenza and RSV did not correlate with SARS-CoV-2 responses [38]. To understand whether the negative association reflects antibody-mediated protection, prospective studies should determine the viral pathogens by PCR to recognize the chronology of infections. The neutralizing capacity of convalescent sera from RSV-infected patients against SARS-CoV-2 virus can also be examined in this regard.

Due to shared clinical manifestations and pathophysiology, co-circulation of dengue and COVID-19 has introduced a major challenge into their diagnosis, treatment, and resource allocation, particularly in regions where dengue is endemic [23]. In this study, we have demonstrated that SARS-CoV-2 seropositivity is higher in dengue-seropositive than -seronegative individuals, at least in the slum population. Serological cross-reaction leading to false-positive serology for SARS-CoV-2 in dengue patients or vice versa has been reported in recent studies [23,39–41], which is also possible in our study. Moreover, to our knowledge, this is the first report to show any association between SARS-COV-2 and chikungunya seroprevalence. Whether there is any mechanistic relationship between dengue- or chikungunya-specific antibodies and SARS-CoV-2 antibodies needs to be addressed in experimental/prospective studies.

COVID-19 patients are prone to suffer from thrombocytopenia (i.e., low platelet count) and the count decreases with increasing disease severity [42,43]. In the present study, SARS-CoV-2-seropositive participants, who mostly experienced mild or no COVID-19 symptoms, had significantly lower platelet counts than seronegative participants. Low platelet counts in these participants probably reflected recent infection with SARS-CoV-2. Moreover, chikungunya-seropositive participants showing lower platelet and higher neutrophil counts compared to seronegative participants, might suggest double burden of recent infections with the two viruses (SARS-CoV-2 and chikungunya). There was no difference in platelet counts between seropositive and seronegative participants for RSV and dengue. This finding may indicate that the infection, particularly with dengue virus, was not recent since dengue infection generally causes a dramatic reduction in platelet counts.

One limitation of the study is that it aimed to perform only a serosurvey of SARS-CoV-2, and detecting infection through antigen or PCR tests was beyond the scope of the study. The time duration between SARS-CoV-2 infection and time of sample collection for serology test would be an important variable to fit into the regression model, since it is known that seropositivity for SARS-CoV-2 fades over time. Another limitation is that influenza A, a major respiratory pathogen was not included in the analysis. There are many serotypes of influenza A, and no data are available to show the serotype(s) prevailing in Bangladesh. Due to budgetary constraints, antibodies to different serotypes of influenza A could not be detected. Our assay determined both IgM and IgG antibodies against SARS-CoV-2, capturing both recent and past infections; however, IgM antibodies were not determined against other viruses, again due to budgetary limitation.

## 5. Conclusions

In summary, the study findings demonstrated a positive association of SARS-CoV-2 seropositivity with antibodies to arboviruses dengue or chikungunya, while a negative association was observed with antibodies to a respiratory virus RSV. Prospective longitudinal studies confirming the infection (by molecular testing) with different circulating viruses should be carried out to evaluate the influence of these viral infections in modulating SARS-CoV-2 infection. In arbovirus-endemic regions, appropriate measures need to be taken to control vector-borne infections and manage the COVID-19 pandemic.

**Author Contributions:** Conceptualization, P.S., E.A., M.Z.I., D.M.E.H., S.A. (Shehlina Ahmed) and R.R.; methodology, P.S., E.A., S.A. (Sharmin Akter), S.R., R.U.K., R.A. and M.J.; formal analysis, P.S. and E.A.; resources, R.R.; writing—original draft preparation, P.S. and R.R.; writing—review and editing, all authors; funding acquisition, R.R. All authors have read and agreed to the published version of the manuscript.

**Funding:** This research was funded by The Foreign, Commonwealth & Development Office (FCDO) through The United Nations Population Fund (UNFPA) (GR-01967), and Global Affairs Canada (GR-01686). icddr,b is also grateful to the Governments of Bangladesh, Canada, Sweden and the UK for providing core/unrestricted support for its operations and research.

**Institutional Review Board Statement:** The study was conducted in accordance with the Declaration of Helsinki, and approved by the Institutional Review Board of icddr,b (PR-20070, dated 28 October 2020).

**Informed Consent Statement:** Written informed consents was obtained from household head and each adult participant of the household, and assent was obtained from 10–17 years old adolescent and consent from their parents.

**Data Availability Statement:** Anonymous participant data and a data dictionary for each variable analyzed in this article, as well as the study protocol, the statistical analysis plan, and the informed consent form will be made available upon requests directed to the corresponding author (protim@icddrb.org). Data can be shared after approval of a proposal through a secure online platform. The institutional review board at the icddr,b will review the proposal and will approve. Additionally, data sharing will depend on the published data access rules of the icddr,b. This will need to sign a standard 385 data access agreement by the icddr,b.

**Acknowledgments:** We gratefully acknowledge all the volunteers who gave consent to participate in this study and spent their valuable time for answering questions and providing samples. We thank all the field and laboratory staff for their significant contribution to this work. Special thanks to Md. Ahsanul Haq for his assistance in data management and statistical analysis.

**Conflicts of Interest:** The authors declare no conflict of interest.

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
