# Peer review of "Antibodies to Commonly Circulating Viral Pathogens Modulate Serological Response to Severe Acute Respiratory Syndrome Coronavirus 2 Infection"

_covid, doi:10.3390/covid2120117_

Round 1
Reviewer 1 Report
The authors recruited over 3 thousands participants and analyzed the serous antibodies to several kinds of viruses including SARS-CoV-2. They found that SARS-CoV-2 seropositivity were higher in dengue and chikungunya seropositive than seronegative participants, and that SARS-CoV-2 seropositivity was lower in RSV seropositive compared to seronegative individuals. They also demonstrated a difference in SARS-CoV-2 seropositivity between inhabitants from the slums and surrounding non-slum areas. I think it a good study with significance. However, I have several concerns about the manuscript.
1. As humans and blood samples were subjected to the analysis of this study, an ethical approval for this research was needed.
2. When analyzing SARS-CoV-2 seropositivity in slum and non-slum participants, I am interested in how the different factors affect SARS-CoV-2 seropositivity and what factor is the most important. More specific analysis may be performed if possible, and deep discussion in needed.
3. The differences of SARS-CoV-2 seropositivity in RSV+ vs RSV- and in dengue/chikungunya+ vs dengue/chikungunya+ are very interesting. These may be due to their different susceptibility or immunity to SARS-CoV-2. To explain this, the authors had better to give some advices to performed more specific clinical analysis or in vivo experiments. This will improve the practicability of the article.
4. The text needs to be carefully checked, as there are some spelling mistakes. The writing need to be improved.
Page 1, Line 22: “SARS CoV-2”, please unify.
Page 3, Line 121, 124: “SARS CoV2”, please unify.
Page 2, Line 56: “Banglades”, please check.
Reviewer 2 Report
Sarter et al. submitted an interesting study investigating the association of SARS-CoV-2 seropositivity with seropositivity to other viruses (Influenza B, RSV, dengue, chikungunya, HCoV-HKU1 and paninfluenza virus) in the slum and surrounding non-slum areas of Dhaka and Chattogram Metropolitan Cities in Bangladesh. They show that seropositivity for RSV was associated with lower odds of seropositivity for SARS-CoV-2, whereas higher odds of seropositivity for SARS-CoV-2 was observed with seropositivity for dengue or chikungunya.
In general, the article is very well written and the methods and results seem robust. There are just three questions remaining:
1. Is there also a difference in other blood cells besides platelets in the study participants?
2. Was the time since SARS-CoV-2 infection recorded and/or fitted into the regression model since it is known that seropositivity for SARS-CoV-2 fades over time?
3. How did the authors make sure that especially the seropositivity for dengue did not influence the assessment of SARS-COV-2 specific antibodies? As there are reports about the cross-reactivity of dengue antibodies in rapid serological tests (Khairunisa SQ et al., Infect Dis Rep. 2021 Jun 7;13(2):540-551. doi: 10.3390/idr13020050; Vanroye F et al., Diagnostics (Basel). 2021 Jun 25;11(7):1163. doi: 10.3390/diagnostics11071163.)
Line 90: it is mentioned that children (10-17 years) participated in the study, however, in later lines (e.g. 152) the authors talk about adolescents? I imagine they mean the same participants?
Round 2
Reviewer 1 Report
No more comment.